# Evaluation and Management of Cavus Foot in Adults: A Narrative Review

**DOI:** 10.3390/jcm11133679

**Published:** 2022-06-26

**Authors:** Boquan Qin, Shizhou Wu, Hui Zhang

**Affiliations:** Department of Orthopedics, Orthopedic Research Institute, West China Hospital, Sichuan University, Chengdu 610041, China; 2021324025286@stu.scu.edu.cn (B.Q.); wushizhou1990@wchscu.cn (S.W.)

**Keywords:** cavus foot, CMT (Charcot Marie Tooth), tendon-transfer, osteotomy, 3D technique, adult, narrative review

## Abstract

Objective: Cavus foot is a deformity defined by the abnormal elevation of the medial arch of the foot and is a common but challenging occurrence for foot and ankle surgeons. In this review, we mainly aim to provide a comprehensive evaluation of the treatment options available for cavus foot correction based on the current research and our experience and to highlight new technologies and future research directions. Methods: Searches on the PubMed and Scopus databases were conducted using the search terms cavus foot, CMT (Charcot–Marie–Tooth), tendon-transfer, osteotomy, and adult. The studies were screened according to the inclusion and exclusion criteria, and the correction of cavus foot was analyzed based on the current research and our own experience. At the same time, 3D models were used to simulate different surgical methods for cavus foot correction. Results: A total of 575 papers were identified and subsequently evaluated based on the title, abstract, and full text. A total of 84 articles were finally included in the review. The deformities involved in cavus foot are complex. Neuromuscular disorders are the main etiologies of cavus foot. Clinical evaluations including biomechanics, etiology, classification, pathophysiology and physical and radiological examinations should be conducted carefully in order to acquire a full understanding of cavus deformities. Soft-tissue release, tendon-transfer, and bony reconstruction are commonly used to correct cavus foot. Surgical plans need to be customized for different patients and usually involve a combination of multiple surgical procedures. A 3D simulation is helpful in that it allows us to gain a more intuitive understanding of various osteotomy methods. Conclusion: The treatment of cavus foot requires us to make personalized operation plans according to different patients based on the comprehensive evaluation of their deformities. A combination of soft-tissue and bony procedures is required. Bony procedures are indispensable for cavus correction. With the promotion of digital orthopedics around the world, we can use computer technology to design and implement cavus foot operations in the future.

## 1. Introduction

Cavus foot is a common problem encountered by foot and ankle surgeons. The term “pes cavus” or “cavus foot” was first used by Shaffer MD in 1885. Cavus foot is defined as a foot with a high medial arch, which has an estimated prevalence of 10% among adults. The latest data on the prevalence of CMT in Norway suggest that its prevalence is 1:1250 [1]. Cavus foot is usually a progressive disease. Flexible cases may only manifest themselves as changes to the shape of the foot following repeated strain. The development of stiffness, pain and even dysfunction in the weight-bearing area may occur, seriously affecting the patient’s quality of life [2].

The arch of the foot plays an important role in walking and running [3]. Since the 1960s, with the emergence of foot pressure testing instruments and gait analyzers, the biomechanics of the foot have been researched in more depth. There are two different phases during the gait cycle: the stance phase and the swing phase [4]. The stance phase accounts for about 60% of the gait cycle, and the swing phase accounts for about 40% of the gait cycle [5]. Both phases are subdivided into different periods (first, second, and third rocker) and events (heel-strike, foot-flat, heel-rise, toe-off, and foot clearance) [6]. Two “heel-strike” events by the same heel delimit one gait cycle. The shape of the arch varies during the different events of a normal gait cycle. The Chopart joint alternates between the “lock” and “unlock” states during the walking process. At the end of the stance phase, the heel is everted and so the calcaneocuboid and talonavicular joints are not parallel and “lock”, becoming immobile to stabilize the midfoot. At the heel-strike and midstance stages, the heel is inverted and thus the two joints are parallel and “unlock”, allowing for motion across the Chopart joint [7]. The first ray and the associated muscle and ligaments play an important role in maintaining the stability and flexibility of the arch. Hypermobility of the first ray occurs during both walking andsports activities and may lead to planus foot or hallux valgus [8,9]. The first ray maintains some flexibility at the beginning of cavus foot, but the persistent plantar flexion of the first ray may occur as it progresses.

The Charcot–Marie–Tooth (CMT) Association published a consensus statement on the surgical treatment of CMT in 2020 [10], which provided a standard for foot and ankle surgeons around the world. Since the etiology and clinical manifestations of cavus foot are variable, surgical plans should be customized to each patient on the basis of a comprehensive understanding of the deformities. Multiple surgical methods have been described. Among them, osteotomy surgeries play an important role, but choosing the appropriate method is often difficult for foot and ankle surgeons. Therefore, we searched the available literature using a certain search strategy (Figure 1). The inclusion criteria included: (1) published in peer-reviewed journals; (2) original studies or reviews on related topics; (3) published in English and (4) full-text. The exclusion criteria included: (1) commentaries; (2) abstracts from conferences; (3) studies on animals and (4) document cannot be downloaded. On the basis of a detailed description of the evaluation methods of cavus deformities, different osteotomy methods were simulated using 3D models to help us gain a more intuitive understanding.

## 2. Biomechanics, Etiology and Classification of Cavus Foot

### 2.1. Biomechanics

The deformities involved in cavus foot are complex and include the abnormal elevation of the medial arch, varus hindfoot, high calcaneal pitch, high-pitched midfoot, plantarflexed, claw toe, and adducted forefoot. The anatomical structure of the cavus foot can be better understood by building a three-dimensional model. (Figure 2). The front view of this model shows the rotation and varus of the forefoot. From the AP (anterior–posterior) view of the model, the adduction and supination of the forefoot will be seen. The TC (talo-calcaneal) angle becomes narrow. From the lateral view, the first ray is in a dramatic plantarflexion position and the fifth ray is overloaded. Viewed from back to front, the varus and pronation of the hindfoot are apparent. Such structural changes seriously affect the flexion and extension of the midfoot by affecting the “lock” and “unlock” mechanism of the Chopart joint, resulting in excessive load on the lateral side of the foot in the varus position. During the gait cycle, the foot remains locked in the forefoot and hindfoot varus position throughout the stance phase, resulting in less stress desorption. These structural changes may lead to metatarsal pain, fifth metatarsal stress fracture, plantar fasciitis, medial longitudinal arch pain, and even iliotibial band syndrome [7].

### 2.2. Etiology

The etiology of cavus foot is most frequently attributed to neuromuscular disorders involving the central or the peripheral nerves [11,12,13,14]. The common causes are listed in Table 1. Two-thirds of adults with symptomatic cavus foot have an underlying neurological condition. Among them, Charcot–Marie–Tooth (CMT) disease, a hereditary sensory motor neuropathy, is most frequently reported. The probability of a patient who has bilateral cavovarus feet being diagnosed with CMT is 78% [15,16]. At the same time, about 34% of children who are diagnosed with poliomyelitis will develop a secondary pes calcaneocavus deformity [17].

### 2.3. Classification

To date, there is no unified classification standard for cavus foot. We usually classify cavus foot according to the causes and physical examination, which has certain guiding significance for us to customize surgical plans.

Causes classification

Neuromuscular;Congenital;Acquired;Idiopathic.

Deformity location classification

Forefoot derived cavus: The apex of the deformity at the midfoot or forefoot, or plantar flexion of the first ray;Hindfoot derived cavus: Rigid varus and inclination increase of the calcaneus;Mixed cavus: A combination of the previous types.

Deformity degree classification [18]

Flexible: Dynamic deformity that corrects with tendon transfers;Stiff: Structural deformity that corrects with soft tissue release and balance;Rigid: Structural deformity that requires osteotomies for correction.

## 3. Pathophysiology of Cavus Foot

The balance of foot muscle strength plays an important role in maintaining foot function. There is a balance between several pairs of important muscles in the foot. For example, the tibialis anterior extends the first ray, while the peroneus longus flexes the first ray; the posterior tibial controls the foot varus, while the peroneus brevis controls the foot valgus; the peroneus longus and the calf triceps flex the foot, while both the tibialis anterior and the extensor longus cause the ankle dorsal extension [19]. Cavus foot typically results from an imbalance between the muscle forces acting upon the foot [19,20,21,22]. However, the pathogenesis of cavus foot is varied. Patients with CMT disease usually begin with progressive weakness of the intrinsic muscle and tibialis anterior, but the strength of the posterior tibial and peroneus longus is normal, which leads to a gradual decrease in the motion of the first ray [23]. The muscle strength of the extensor hallucis longus increases to compensate for the weakness of the tibialis anterior, which further aggravates the flexion of the first ray. As the disease progresses, the first ray continuously worsens and this results in forefoot external rotation and metatarsal adduction. The deformed forefoot is flexible at first, and patients can keep the plantigrade foot. As the deformity of the forefoot becomes stiff, the hindfoot begins to develop compensatory varus, which leads to plantar fascia and Achilles tendon contracture. The weakness of the intrinsic muscle aggravates the flexion of the interphalangeal joint and overextension of the metatarsophalangeal joint, resulting in atrophy of the fat pad at the metatarsal head and the formation of corpus callosum and claw toes (Figure 3). The cavus foot deformity caused by poliomyelitis sequelae develops through other mechanisms [24]. The muscle strength of the tibialis anterior and intrinsic muscle of the foot is usually normal, while the strength of the tibialis posterior and calf triceps is weak. These kinds of patients usually have a typical pes calcaneocavus deformity (Figure 4). In patients with spina bifida and tethered cord syndrome, muscle strength imbalance is caused by nerve damage that dominates the muscles.

## 4. Clinical Evaluation

### 4.1. Physical Examination

Detailed medical history inquiries and physical examinations are very important for evaluating the condition of patients with cavus foot. We should find out about the patient’s date of birth, growth and development, family history, and medical history in detail. It can be helpful to examine the feet of both parents. It is not unusual to find that one of the parents has a cavus foot as well. Just as the cause of a cavus foot can vary, so can the clinical presentation. Typical clinical presentation of cavus foot includes unstable gait, ankle pain, plantar painful callus, loss of sensation in the metatarsal head, and occasional knee and hip pain. Patients with CMT disease may also present hand deformities and thenar muscle and interosseous muscle atrophy (Table 2).

A thorough, top–down physical examination is necessary for the evaluation of cavus foot, both to identify the source of the deformity and to guide the surgical plan. First of all, a visual inspection of the back should be performed, looking for scoliosis or spina bifida, either of which may point to a spinal etiology of the cavus foot. Then, check whether there are deformities in the hip and knee joints and whether the range of motion is normal. Finally, have the patient sit in a chair facing the doctor, put the patient’s foot on the examiner’s leg, and make it as relaxed as possible.

Foot examination should include the following aspects:A short assessment of the patient’s gait. You may be able to see an unstable foot drop gait, a weak push stance, or a weak dorsiflexion. Patients who have limb length discrepancies may have a limp gait. Patients with hip pathologies may also present with a Trendelenburg gait.Examine the appearance of the feet and toes. Check whether the feet are symmetrical, whether there are claw-toe deformities, and whether the range of motion of the forefoot, hindfoot, ankle, and toes are normal. Check whether the “peek-a-boo” sign [25,26] is obvious.Identify the deforming forces involved [27]. Although we cannot precisely calculate muscle strength through a physical examination, it is of great significance to help us to formulate the correct tendon release or transfer plan.Identify the source of the deformity. The Coleman block test [28] is the most popular method for assessing the flexibility of the hindfoot and for determining the source of the deformity. The patient is asked to stand with the heel and lateral border of the foot over a 1-inch-high block while the medial metatarsals are in contact with the floor (Figure 5). When the hindfoot is flexible, the heel will return to a neutral or valgus position. Meanwhile, we can further judge that the source of the deformity is the forefoot. A positive Coleman block sign implies that the hindfoot varus is due to the plantarflexed first ray and that the hindfoot is flexible. However, the Coleman block test cannot accurately reflect the flexibility of the hindfoot, especially when the patient has difficulty positing their foot on the block, which will mislead our judgment. Price et al. [29] described a method of evaluating the flexibility of the hindfoot. The patient lies prone on the examination bed, the examiner instructs the patient to bend the knee and corrects the calcaneal varus manually to see if the varus can be completely corrected and whether the first ray is decreased. Therefore, we need to make a comprehensive judgment by combining manual correction with the Coleman block test. If manual correction of the varus of the hindfoot is possible, we can also think that the hindfoot is flexible. More flexion of the first ray is usually observed after manual correction of the forefoot pronation and hindfoot varus (Figure 6).Assess the deformity in the coronal plane. The three-dimensional nature of cavus foot requires us to take into account all aspects during correction. We should estimate whether there is adduction or abduction of the forefoot after the manual correction of the hindfoot.Assess the mobility of the first ray. The first metatarsal of the patient’s foot is grasped with the examiner’s hand and the rest of the foot is stabilized. The first metatarsal is then manipulated with the right thumb and forefinger. Motion in multiple directions should be evaluated [30]. In cases of cavus foot, the motion of the first ray is usually decreased.The Silfverskiold test [31,32,33] is essential. The clinician places the one hand at the level of the subtalar joint and the other around the midfoot, stabilizing the talonavicular joint and keeping the foot in a neutral position while dorsiflexing the ankle. The test is performed with the knee extended and flexed, respectively. If ankle dorsiflexion is not affected by knee flexion and extension, this indicates Achilles tendon contracture. If ankle dorsiflexion is increased during knee flexion, this indicates gastrocnemius contracture. We can choose gastrocnemius resection or Achilles tendon lengthening according to the result of this test.

### 4.2. Radiological Evaluation

Radiological evaluation should be emphasized, as simple radiographs can aid considerably in operative decision making [34,35]. Radiographs should be taken with the patient’s weight-bearing on the foot and ankle so as to allow for the assessment of the true deformity. Radiographs of the foot including anteroposterior (AP), oblique, and lateral views are important to identify the apex of the deformity, the latter in particular. On the AP view we may observe a decreased AP–talo–first metatarsal (TM1) angle and TC (talo-calcaneal) angle. On the lateral view, an increased lateral talo–first metatarsal (Meary) angle, decreased talar declination, increased calcaneal pitch (Pitch) angle, and a decreased lateral talo-calcaneal (Djian–Annonier) angle may be seen. The apex of the deformity is always proximal to the base of the first metatarsal, either at the first tarsometatarsal (TMT) joint, the cuneiform, the naviculocuneiform (NC), or the talonavicular (TN) joint. In addition, AP, mortise, and lateral views of the ankle should be obtained to identify whether arthritis or varus exists. The long axial X-ray of the calcaneus allows for the assessment of the hindfoot alignment in the coronal plane, which is usually the varus in cavus foot. The tibial calcaneal angle (TCA) may be increased (Figure 7). The full-length X-ray of lower limbs in weight-bearing is taken to evaluate the alignment of the lower extremities and observe the presence of knock knee or tibial rotation. The radiological evaluation of cavus foot should also include the Coleman block test in order to assess the flexibility of the foot. In the recent literature, Dr. Michalski [36] and Dr. Tonya An [37] have conducted high quality research about bone morphology changes in cavus foot. Their research results indicate that the TN coverage angle (Figure 7) plays an important role in evaluating the effect of cavus foot correction.

While standard radiographs can provide us with sufficient information, advanced imaging techniques, such as tomography (CT) and magnetic resonance imaging (MRI), can enable us to conduct a more in-depth analysis of cavus foot. Three-dimensional (3D) CT scans show the deformity more intuitively, which is helpful to clinicians to obtain a more comprehensive grasp of the anatomy of cavus foot. MRI may be valuable for assessing unsteady lateral ligaments, identifying peroneal tendon tears, noting the presence of any osteochondral lesions, and demonstrating soft tissue or bone infection.

Gait analysis is used to evaluate the posture changes in the whole body caused by cavus during walking, which has a certain guiding significance for choosing a tendon transfer plan (Figure 8).

In cases of a suspected neuromuscular disorder, an electromyogram should be carried out in order to give better insights into the strength of the muscles and the degree of neuropathy.

## 5. Management of Cavus Foot

### 5.1. Conservative Treatment

Nonoperative treatment of both neuromuscular and non-neuromuscular cavus foot should be tried first in order to prevent or slow down the further development of the deformity, although in many cases the effect is often not satisfactory. Since cavus foot is usually progressive, the deformity is already severe when the symptoms begin to appear. Patients should be treated with medical and physical therapy, including activity modification, anti-inflammatory medications, and shoe modification, which helps to maintain the flexibility of the arch of the foot [38,39,40,41]. Although there are no high-level studies proving that conservative treatment is effective, it is still reasonable to undergo nonoperative treatment in an attempt to avoid operative intervention. The surgical indications of cavus foot mainly include pain, progressive deformity, and unstable gait.

### 5.2. Surgical Treatment

The objective of a successful operative treatment is to achieve a painless, plantigrade, mobile foot [42,43]. Therefore, the key to making a surgical plan for the correction of cavus foot is to understand that anatomic findings do not indicate the cause of any particular component of cavus foot. A personalized operation plan needs to be made according to the results of physical examinations and auxiliary inspections.

At present, the main surgical methods include soft tissue balance, osteotomy, arthrodesis, and Ilizarov external fixation correction. Complex deformities always require the combination of a variety of surgical methods for multiple operations. The principle of surgical treatment requires us not only to correct the existing deformities and restore the plantigrade foot, but also to retain the joints of the foot as far as possible, delay joint degeneration, and reduce the recurrence rate.

For cavus foot caused by stroke, brain trauma, and other diseases with a short course, there are only muscle strength imbalances and soft tissue contractions without bony deformations. In these cases, operations should focus on the reconstruction of foot muscle balance, but such cases are relatively rare in clinics. Almost all patients have long-term diseases and develop stiff or rigid bones, which can only be completely corrected by a combination of osteotomies, tendon transfers, and ancillary procedures (Table 3).

#### 5.2.1. Soft-Tissue Release

It is critical to have adequate soft tissue release before correcting structural deformities. In almost all prolonged cavovarus cases, Achilles tendon tightening and medial and plantar tissue contraction may occur. Start surgery with an Achilles or gastrocnemius to take the varus and pull off the calcaneus. Percutaneous triple hemisection [44] or open Z-plasty of the Achilles tendon [45], and gastrocnemius recession [46] can be selectively performed according to the result of the Silfverskiold test. Then, conduct all soft tissue releases, including spring ligament and midfoot joints. Only then does the surgeon know what osteotomies have to be conducted. Correct the heel first, if it is needed, and then move forward.

Persistent plantar fascia (PF) contracture increases the rigidity of deformity, resulting in foot shortening and stiffness, which needs to be fully released during the operation to reduce the deformity and the amount of osteotomy. We suggest dividing the PF in the midfoot, which has the most power in freeing up a severely depressed first metatarsal. Steindler surgery [47,48] is another method for releasing PF. However, due to its limited release effect and high complication rate, it is rarely used nowadays. In severe cases, the abductor hallucis fascia may also require release, which can be performed through the same incision. Attention should be paid to avoid damaging the blood vessels and nerves of the plantar tissue.

When the residual deformity is still severe after Achilles tendon lengthening and plantar fascia release, additional release of the posterior tibialis tendon, deltoid ligament, and flexor hallucis longus [49] is also necessary.

The deformity should be reevaluated in order to make minor revisions to the bony reconstruction plan after full soft-tissue release.

#### 5.2.2. Bony Reconstruction

Bony reconstruction is mainly achieved by osteotomy or arthrodesis at the apex of the deformity. Before the operation, the X-ray film should be carefully measured and the osteotomy angle should be calculated. The plan may be adjusted according to the actual situation during the operation.

##### Forefoot Driven Cavus Foot

The forefoot-driven cavus foot, when the apex of deformity is around the first TMT joint, can be corrected through a first metatarsal osteotomy [50,51,52,53,54,55,56]. The osteotomy is carried out at a point that is 10 mm distally from the first TMT joint, which can be a dorsal closed wedge osteotomy or a vertical osteotomy translation of the metatarsal shaft (Figure 9). A dorsal wedge is removed and this raises the first ray to correct the flexion deformity. In the process of correction, attention should be paid to avoiding excessive shortening or rotation with subsequent transfer metatarsalgia. The vertical metatarsal osteotomy can prevent metatarsal shortening, but it must be firmly fixed by a hook or step plate. However, we found that the arch of the foot cannot be corrected by osteotomy of the first metatarsal bone alone, which results in residual deformity since the apex of deformity in most cases is usually located in the cuneiforms or more proximal levels.

If the deformity is caused by the plantar-flexing of multiple metatarsals, a Jahss tarsometatarsal truncated-wedge arthrodesis [57] may be indicated (Figure 10). The procedure is essentially a Lisfranc joint arthrodesis. The operation method of the Jahss operation is to perform a distal osteotomy from the base of the first metatarsal from inside to outside along the direction of the Lisfranc joint, usually up to the base of the fifth metatarsal to form a curved osteotomy line. The proximal osteotomy line is usually curved from the medial cuneiform to the cuboid, which is consistent with the curve of the first osteotomy line. The angle and position of the osteotomy should be determined according to the degree of deformity of the patient. The base of the wedge is wider at the dorsal aspect compared with the plantar, and the forefoot is elevated out of its plantarflexed attitude. Usually, the osteotomy at the second and third metatarsal is larger than that of other metatarsals. After the osteotomy is completed, the forefoot is raised. Meanwhile, the rotation or adduction of the forefoot is corrected by adjusting the arthrodesis position. Excessive removal of metatarsals or cuneiform will result in scaphoid foot deformity.

In order to achieve a total correction at the apex of the deformity, it is necessary to consider the midfoot osteotomies, such as the Cole osteotomy, the Japas osteotomy, the Akron osteotomy, or the Myerson osteotomy [27,58,59].

The Cole osteotomy was first proposed by Saunders in 1935 [60] and popularized by Cole in 1940 [61]. The correction is made via a dorsal closing wedge osteotomy, with one cut through the navicular and cuboid and the other cut through the cuneiforms and cuboid. When the osteotomy is completed, the dorsal wedge is removed and the forefoot is raised (Figure 11). In this process, it is important to retain the Chopart joint. In addition, the osteotomy line can be adjusted according to the deformity to achieve multi-plane correction. In 2004, Brandon L. Tullis et al. [62] retrospectively analyzed the clinical data of eight patients (11 feet) who underwent a Cole midfoot osteotomy in Pennsylvania Western Hospital from February 1998 to October 2000. The authors found that all patients achieved good imaging and clinical results and returned to normal walking. There were no obvious complications. In 2019, Ergun S et al. [63] reported on six feet in five patients with cavus foot caused by different causes and treated with a Cole osteotomy. The AOFAS score, imaging findings, and clinical manifestations were significantly improved after the operation. However, no large-cohort studies have been published to date. In clinical work, we find that when the deformity angle is too large, a Cole osteotomy may be not applicable, as it leads to the smaller foot being shorter. However, how to define the scope of the osteotomy angle requires further research.

Japas [64] introduced a dorsal V-shaped osteotomy for the correction of cavus foot in 1968. The intersection of the two osteotomy lines (that is, the V-shaped vertex) is usually located at the navicular, and both sides extend distally through the cuboid and the medial cuneiform, respectively. The V-shaped angle is determined according to the degree of deformity. After the osteotomy is completed, the distal end of the osteotomy is pressed and the metatarsal head is lifted, lowering the arch of the foot in the process. The correction of adduction or abduction can be accomplished by the simple manipulation of the forefoot (Figure 12). Then, the osteotomy plane is fixed with two crossed K-wires after the correction is completed. In 2002, Giannini et al. [65] reported on 69 feet in 39 patients with forefoot-driven cavovarus. All patients received the Japas osteotomy, and 14 feet underwent ancillary procedures. The hindfoot alignment, the Maryland Foot Score, and gait analysis were significantly improved post-operation. In 2009, Protyush Chatterjee [66] reported that the Japas osteotomy was used to correct cavus foot caused by poliomyelitis. The clinical data on 18 cases from 1995 to 2005 were analyzed retrospectively, which concluded that the Japas osteotomy was an ideal choice for adolescent cavus foot correction. The chief advantage of the Japas osteotomy is its widening of the foot in the mid-tarsal region. Joint sacrifices are also inevitable.

Wilcox and Weiner [67] first introduced the “dome” osteotomy method of the midfoot to correct cavus foot deformity in 1985, which is also known as the Akron osteotomy. Two roughly parallel dome-shaped osteotomy cuts are conducted at the apex of the deformity across the midfoot that is wider in the dorsal than the plantar to allow for the raising of the forefoot out of the equinus and the lowering of the longitudinal arch. Appropriate wedging of the osteotomy cuts is then used to accommodate any directional alteration for varus, valgus, dorsiflexion, plantar flexion, or rotation (Figure 13). Once the adequate correction is achieved, the foot is maintained in position by two crossed K-wires. They followed up 22 patients (35 feet) with cavus foot treated by Akron osteotomy for an average of 3.4 years. The results showed that the deformities of all patients were corrected, the plantigrade foot was restored, and the pain was significantly alleviated post-operation, with a satisfaction rate of 67%. Among the patients over 8 years old, the satisfaction rate was 94%. In 2008, Weiner et al. [68] retrospectively analyzed the clinical data of 89 patients (139 feet) who were treated with an Akron osteotomy for cavus foot from 1972 to 2001. By comparing the appearance and pain release of the foot pre- and post-operation, the results showed that all patients achieved perfect deformity correction and had no weight-bearing area pain. The satisfaction rate was 76%. Like the Cole osteotomy, the Akron osteotomy also leads to foot shortening, and the shape of the osteotomy fragment is difficult to control and requires accurate preoperative design and intraoperative operation.

Myerson [27] proposed a new joint-sparing midfoot osteotomy method in 2013, the modified Japas osteotomy. The osteotomy is performed between the medial, middle, and lateral cuneiform. The first osteotomy line is vertical and the second and third osteotomy lines are oblique from the medial cuneiform and cuboid to the intersection of the middle cuneiform (Figure 14). The amount of osteotomy of the medial cuneiform and cuboid is adjusted according to the degree of deformity. The varus, flexion, and adduction of the forefoot are corrected through a combination of dorsiflexion, elevation, and rotation, which are completed after the osteotomy.

Despite the variety of midfoot osteotomy methods described above, the Cole is by far the best and most widely accepted all over the world [10].

The Ilizarov method [69,70] of external fixation can be combined with a midfoot osteotomy to correct complex foot deformities due to its three-dimensional correction property.

This method involves minimally invasive surgery, and as such the risk of soft tissue and neurovascular damage is much lower. It allows for dynamic three-dimensional correction and can simultaneously correct other associated lower extremity deformities. Furthermore, early weight-bearing is beneficial to the recovery of bone mass and the healing of the osteotomy area.

##### Hindfoot Driven Cavus Foot

If the hindfoot varus is fully corrected with the first metatarsal or midfoot osteotomy, a calcaneus osteotomy is not necessary. However, if there is residual varus after the dorsiflexion osteotomy, a calcaneus osteotomy should be carried out.

Multiple techniques for calcaneus osteotomy have been described, but making the correct choice is difficult for many orthopedists. We need to choose the appropriate type of osteotomy according to the extent of the varus of the hindfoot [71].

For a mild varus, a Dwyer [72] lateral closing wedge osteotomy (Figure 15) may be sufficient. For a moderate varus, a lateral translational osteotomy [73] (Figure 16) is more suitable. It can be combined with a Dwyer osteotomy to achieve a three-dimensional correction by rotating, translating, and elevating the posterior fragment. For a severe varus, a Z-shaped osteotomy [74] (Figure 17) is a powerful tool. This osteotomy can also provide three-plane correction, and the ability to correct the calcaneal varus is stronger. Z-shaped osteotomies are technically challenging and require full exposure. Attention should be paid to protecting the lateral and the medial neurovascular structures. Meanwhile, overcorrection should be avoided.

Dr. Glenn B. Pfeffer et.al. [75] used 3D prints to compare the efficacy of three different calcaneal osteotomies for the correction of heel varus. Different groups were designed and osteotomies were performed in different ways. Measurements were calculated using multiplanar reconstruction image processing, and the results were statistically analyzed. After detailed comparative studies, they found that Dwyer, oblique, and Z osteotomies did not create either lateral translation or coronal rotation without the addition of a lateralizing slide or rotation of the posterior tuberosity. On this basis, they proposed that Dwyer and oblique osteotomies would be best suited for mild deformities and Z-shaped osteotomy was suitable for severe ones. This study is important to consider when choosing the correct method for heel varus correction. Patients with rigid and severe cavus foot deformity complicated with osteoarthritis can only be treated using arthrodesis in order to restore the plantigrade foot. A triple arthrodesis [76] is carried out on the basis of complete correction of cavus foot deformity. The rotation of the forefoot needs to be corrected, the arch of the foot is reduced to normal, and the hindfoot is placed in a mild valgus position. It is useful to flat cut the calcaneocuboid joint with a saw to avoid a painful bony bump upon weight-bearing.

##### Mixed Cavus Foot

For the mixed cavus foot, the forefoot and rear foot should be corrected at the same time according to the previous methods. It is usually corrected in the order from back to front. Some patients’ deformities are severe and rigid, and in these cases, arthrodesis should be performed.

#### 5.2.3. Soft-Tissue Balancing

Muscle imbalance plays an important role in the occurrence and development of cavus foot [77]. Thus, soft-tissue balance should be carried out after bony structure reconstruction. Without well-balanced muscle power, the deformity will recur and the correction will eventually fail. A detailed soft-tissue balancing plan should be drawn up according to manual muscle testing. The tendon chosen to transfer should have strength no less than grade 4 or 5 [19]. The deforming force usually includes the peroneus longus tendon, the posterior tibial tendon, and the equinus.

The main goals of tendon transfer include: (1) improve motor function for restoration of balance and prevention of contractures; (2) eliminate deforming forces and (3) stabilize the alignment of the foot.

Peroneus longus transfer to brevis is a simple and effective method to aid reconstruction [78]. Overactivity of the peroneus longus contributes to the persistent plantar flexion of the first ray, resulting in a forefoot-driven cavus. Suturing the peroneus longus and brevis together will augment the valgus strength and eliminate the deforming force to prevent a recurrence. If the peroneus brevis tendon is torn or degenerated, the diseased part should be excised. If it is torn to a serious degree, the peroneus longus can be fixed to the base of the fifth metatarsal.

When the posterior tibial tendon overpowers the peroneus brevis, it will result in foot inversion, heel varus, and the contracture of the medial soft tissues. When the patient has moderate varus deformity, the posterior tibial tendon is anchored into the middle or lateral cuneiform to compensate for the dorsiflexion force; when the patient has severe varus deformity, the posterior tibial tendon is anchored into the cuboid to compensate for the force of dorsiflexion and eversion. In milder cases without tibialis anterior weakness, the anterior tibial tendon can be transferred to the middle cuneiform to reduce the supination force while augmenting the dorsiflexion power [23].

Other tendon transfer plans should be applied depending on individual conditions. The options include the split posterior tibial tendon lateral transfer, transfer of the flexor digitorum longus to the peroneus brevis, the extensor digitorum longus transfer to the anterior tibialis, and so on.

#### 5.2.4. Ancillary Procedures

Hallux hammertoe deformity is a frequent component of cavus foot deformity, which is the result of the imbalance between the flexor and extensor muscles of the great toe. The Jones procedure [79,80] releases the extensor hallucis longus from the distal phalangeal and inserts it into the neck of the first metatarsal. The distal interphalangeal joint is fused with a screw.

Hibbs tendon transfer [81] is a transfer of the extensor digitorum longus to the lateral cuneiform in order to correct the deformity of claw toes and aid in dorsiflexion.

Cavus foot is usually accompanied by ankle instability, so the repair of the lateral collateral ligament will be selectively performed.

## 6. What’s New

With the development of computer technology, many orthopedic surgeons have begun to use computer-aided surgical planning and 3D-printed osteotomy guide plates to assist in cavus foot correction. Meanwhile, 3D printing technology has also been used to assess the orthopedic results (Figure 18). Dr. Louis Dagneaux et al. [82] from France first reported the technique of a 3D-printed patient-specific cutting guide for anterior midfoot tarsectomy in 2020. In their article, they state that due to the multiplanar deformed nature of cavuvarus, midfoot osteotomies should be performed in some cases. However, this procedure remains difficult to perform, and the margin for error is small. Therefore, they used 3D software to design osteotomy lines based on the principles of the Cole midfoot osteotomy and allowed all imaging angles to return to the normal range. On this basis, a cutting guide was generated and verified by computer simulation. The accuracy and convenience of this cutting guide were verified during surgery. Dr. Francisco B. Sobrón et al. [83] from Spain, reported the technique tip of the 3D printing surgical guide for pes cavus midfoot osteotomy in 2021. The design process was similar to that of Louis Dagneaux et al. This surgical guide was used for correcting cavovarus in three patients and achieved good results. They found that this method could minimize human error and improve accuracy while reducing operation time and intraoperative X-ray exposure. Dr. Glenn Pfeffer et al. [75] from America reported on using 3D printing technology to compare the effects of different calcaneal osteotomy methods (Dwyer, oblique, and Z osteotomies) on the correction of heel varus in cavovarus patients. They came to the conclusion that these three methods did not create either lateral translation or coronal rotation without the addition of a lateralizing slide or rotation of the posterior tuberosity, which has important clinical significance for the correction of heel varus in cavus foot. We have also tried to use 3D-printed, customized guides for cavus foot correction recently. Good clinical results have been achieved (relevant data are being collected and collated).

## 7. Discussion

The evaluation and management of cavus foot are based on a comprehensive understanding of this deformity. It is important to look for the underlying causes, as some patients are not candidates for surgical treatment. For example, the correction of a hypertonic deformity caused by spastic cerebral palsy may lead to myasthenia paralysis [19].

The correction of cavus foot can be difficult with regard to both the decision-making and the execution of the surgery [42]. The correct evaluation of the cavus deformity is the key to choosing the correct treatment. In order to achieve this, we should conduct a series of evaluations, such as physical examination, radiographic scans, gait analysis, and so on. On this basis, surgical planning can be carried out. The correct selection of surgical methods plays a decisive role in the prognosis of patients. As described above, any bony correction must be in conjunction with the soft-tissue releasing and balancing procedure [12]. The soft-tissue procedures are customized according to the muscle strength and stiffness degree of the foot. We also need to pay attention to the mobility of the first ray. The restoration of arch height and normal range of motion of the first ray by osteotomy, tendon transfer, and soft tissue release is essential. However, the overcorrecting of deformities should be avoided as it may lead to the hypermobility of the first ray, the collapse of the arch, or hallux valgus [8].

Here, we focus on bony procedures for deformity correction. We have simulated a variety of osteotomy methods using 3D models. Among them, the first metatarsal osteotomy was calcaneus osteotomy, which was widely used for cavus foot correction in past decades and achieved good clinical effects [52,54,55]. Dr. Glenn B. Pfeffer and his team [75,84] have carried out high-quality research on calcaneus osteotomy, including its orthopedic effects and biomechanical evaluation. However, when the apex of the cavus foot deformity is located in the midfoot, the correction in the apex of the osteotomy is unavoidable. A variety of midfoot osteotomy methods have been described in detail in the previous section, among which the Cole, Japas, and Akron have been repeatedly reported for cavus foot correction [58,59,62,63,66,68]. The Cole is by far the most widely accepted all over the world [10]. However, due to the multi-plane correction of the Cole osteotomy, there are high requirements for preoperative design and intraoperative operation. Due to this, some scholars have explored the use of 3D printing technology to assist with the Cole osteotomy. Three-dimensional printing technology has been used to simulate pre-operative osteotomies, and intraoperative cutting guide-assisted osteotomy has been used to achieve accurate and efficient surgical results [82,83]. At present, we are conducting research on this and are hoping to design relevant surgical instruments based on 3D simulation in order to realize AI diagnosis and even robotic surgery.

The limitations of our research are as follows: our paper is a narrative review, which combines current research with our own experience. There are problems, such as low levels of evidence and no systematic evaluation, so a quantitative analysis cannot be conducted. In future studies, we will conduct a systematic review or meta-analysis on a specific aspect of cavus foot correction and conduct a quantitative analysis on related issues. We hope to provide high-quality research results from our clinical work.

## 8. Conclusions

The treatment of cavus foot requires personalized operation plans according to different patients based on the comprehensive evaluation of the deformity. A combination of soft-tissue and bony procedures is required. Bony procedures are indispensable for cavus correction, but many bony procedures are difficult to conduct and have long learning curves.

With the promotion of digital orthopedics around the world, we can use computer technology to design and implement cavus foot operations in the future. Meanwhile, we hope that high-level studies in the field of cavus foot correction can be published as soon as possible to guide our treatment strategies.

## Figures and Tables

**Figure 1 jcm-11-03679-f001:**
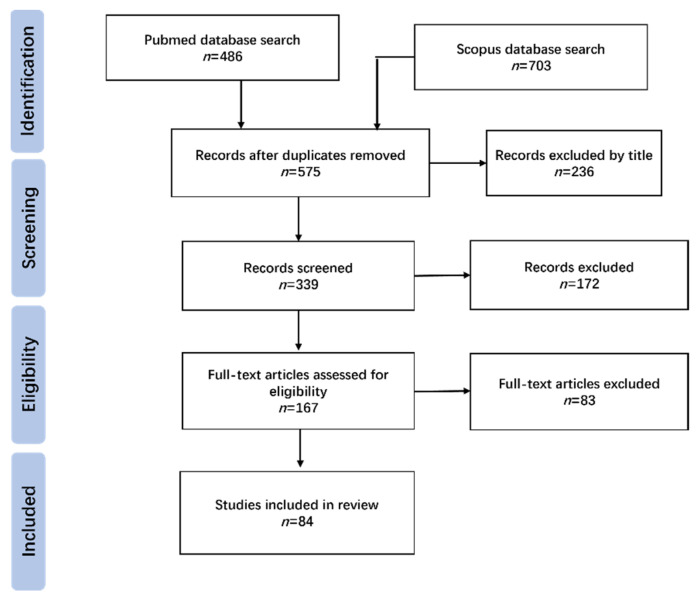
Flow diagram of literatures review.

**Figure 2 jcm-11-03679-f002:**
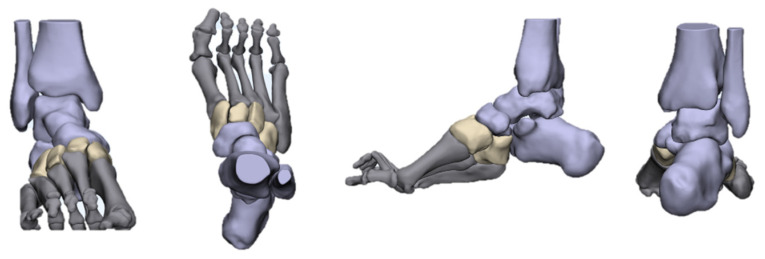
Three-dimensional view of cavus foot.

**Figure 3 jcm-11-03679-f003:**
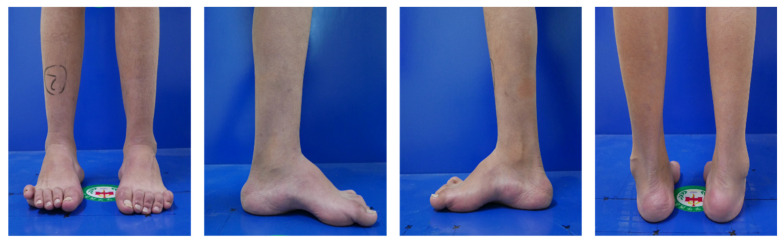
The outlook of bilateral cavovarus foot.

**Figure 4 jcm-11-03679-f004:**
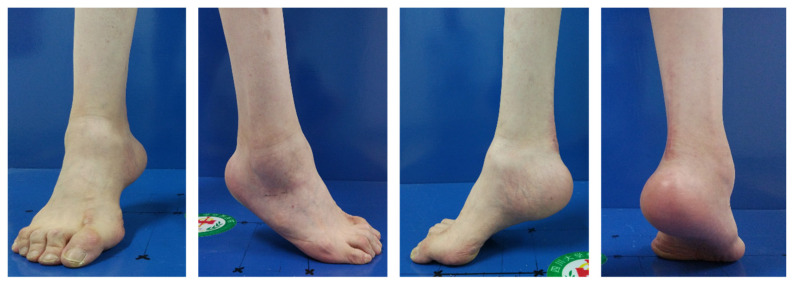
The outlook of unilateral pes calcaneocavus deformity.

**Figure 5 jcm-11-03679-f005:**
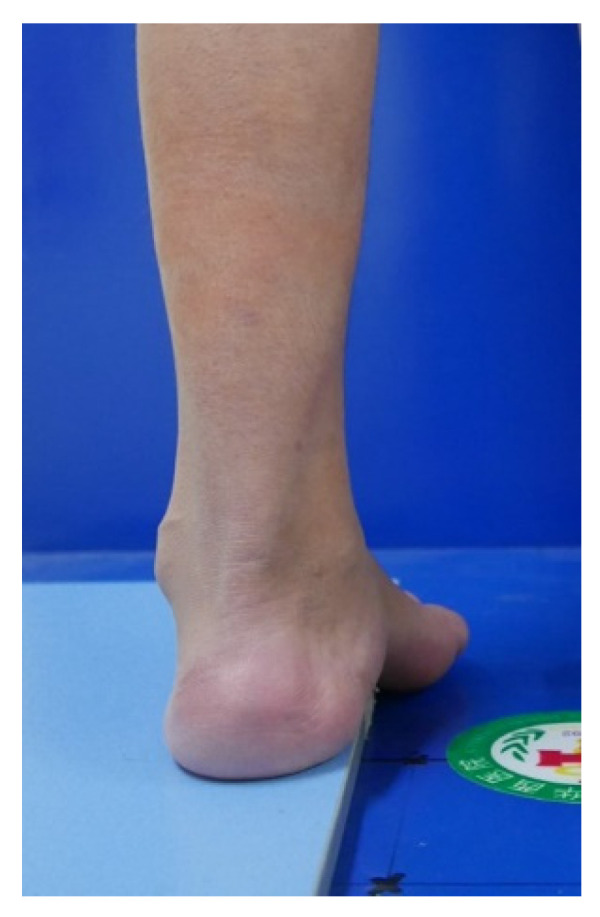
The Coleman Block Test (A test for evaluating the flexiblity of the hindfoot).

**Figure 6 jcm-11-03679-f006:**
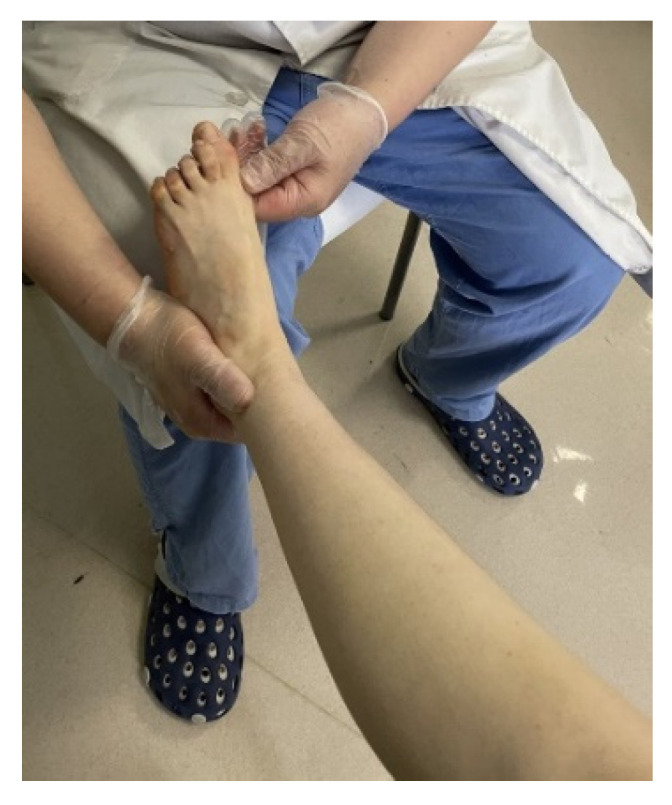
Manual correction of the varus of hindfoot (The first-ray becomes more flexion).

**Figure 7 jcm-11-03679-f007:**
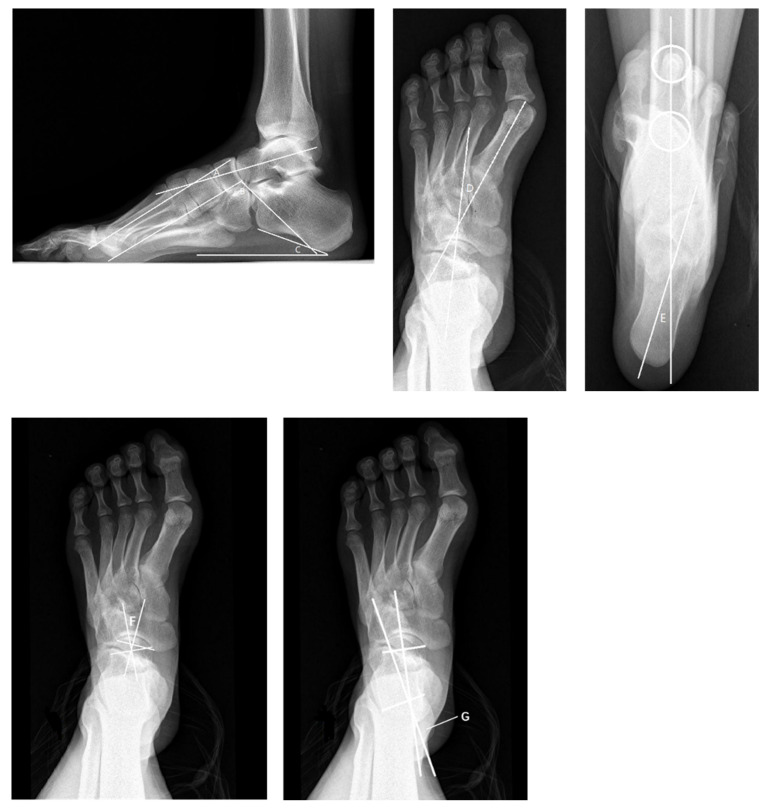
The radiological evaluation of cavus foot pre-operation. A: Meary Angle (−4°~4°) *. B: Djian–Annonier Angle (120°~130°) * C: Pitch Angle (20°~30°) * D: TM1 Angle (0°~20°) * E: TCA Angle (0°~5°) *. F: TN Coverage Angle (1.8°~19.3° male 6.7°~21.7° female) *. G: TC Angle (15°~35°) *. * Represent the normal range.

**Figure 8 jcm-11-03679-f008:**
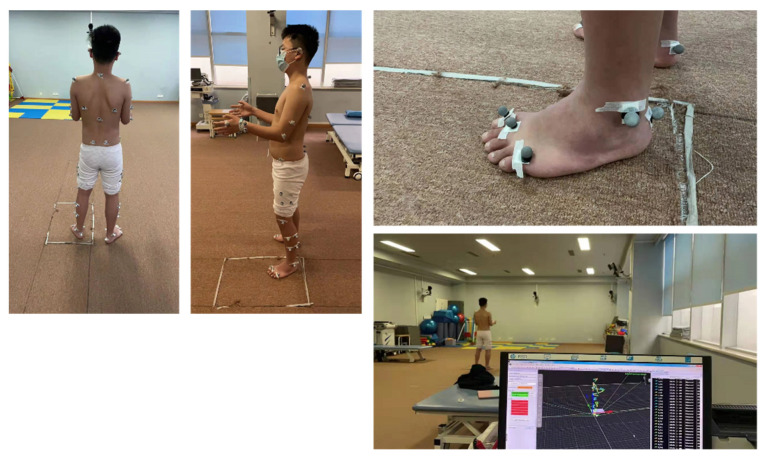
The 3D gait analysis of cavus foot pre-operation.

**Figure 9 jcm-11-03679-f009:**
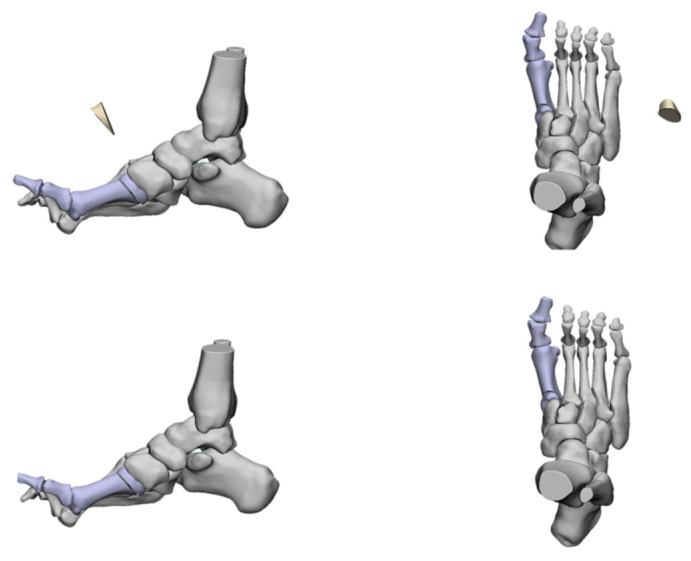
The dorsal closed wedge osteotomy of the first metatarsal.

**Figure 10 jcm-11-03679-f010:**
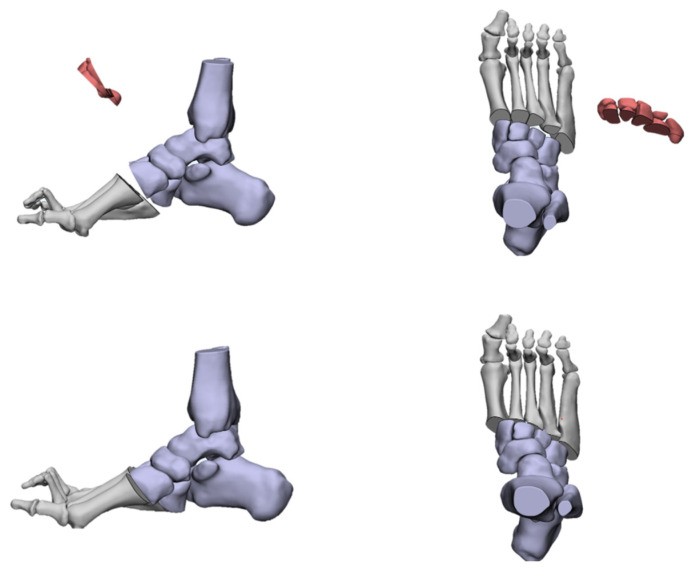
The Jahss Osteotomy.

**Figure 11 jcm-11-03679-f011:**
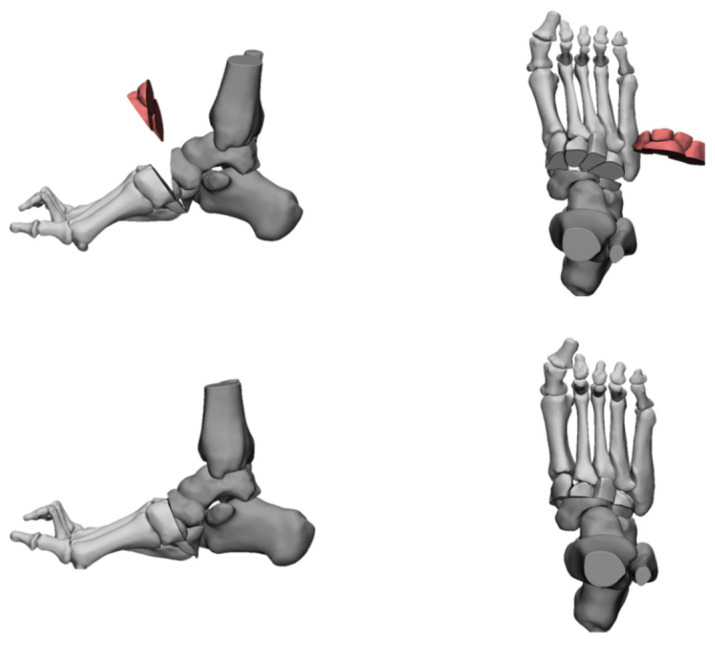
The Cole Osteotomy.

**Figure 12 jcm-11-03679-f012:**
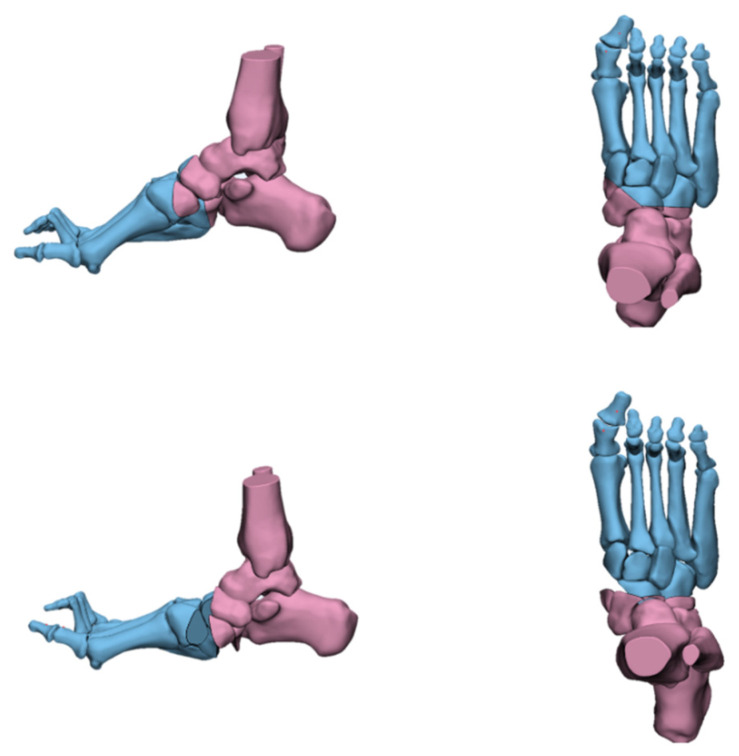
The Japas Osteotomy.

**Figure 13 jcm-11-03679-f013:**
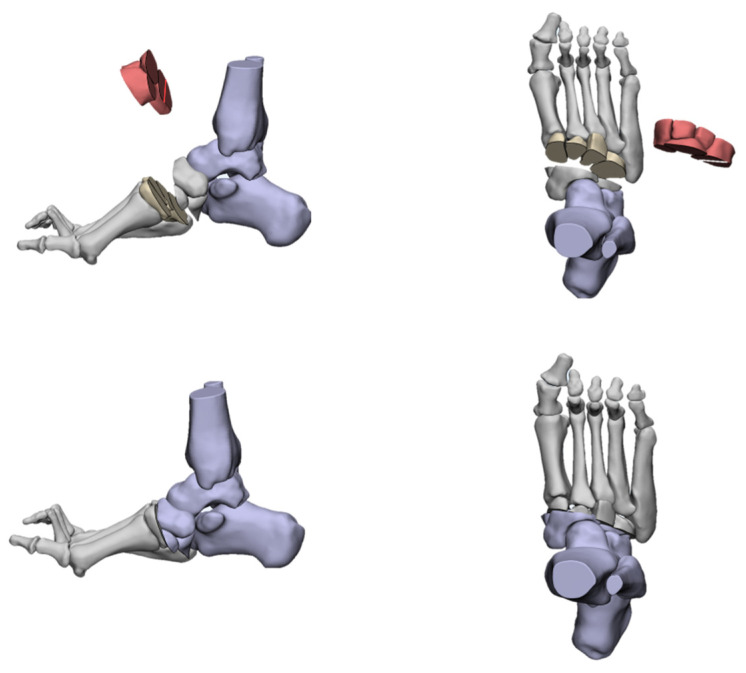
The Akron Osteotomy.

**Figure 14 jcm-11-03679-f014:**
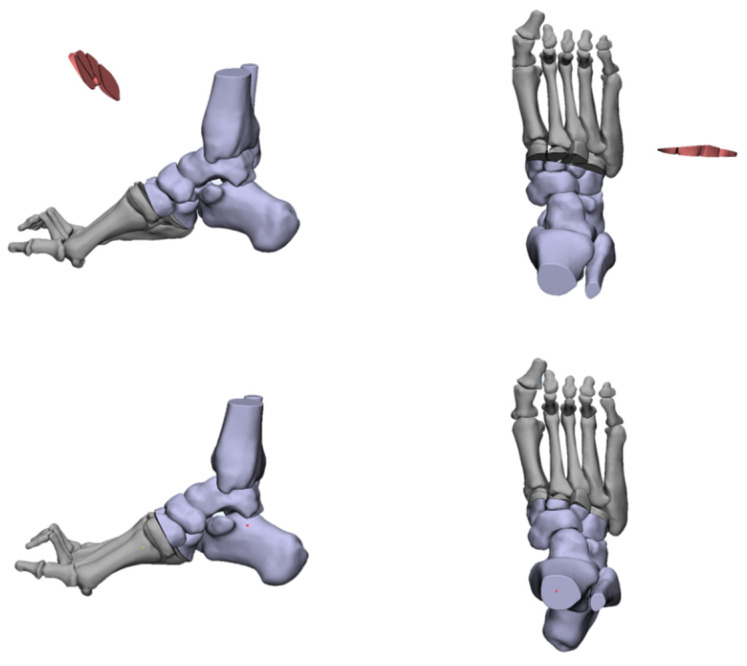
The Myerson Osteotomy.

**Figure 15 jcm-11-03679-f015:**
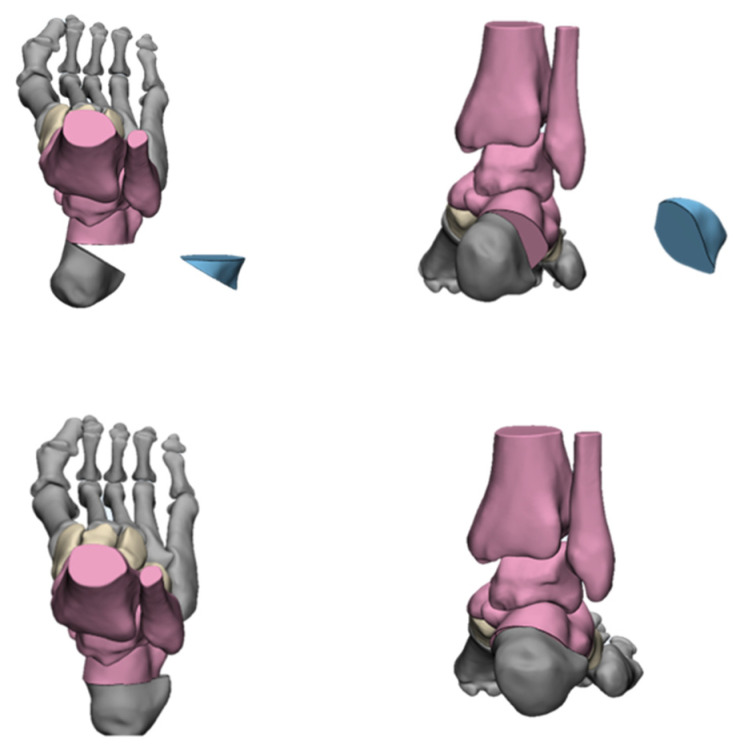
The Dywer calcaneus Osteotomy.

**Figure 16 jcm-11-03679-f016:**
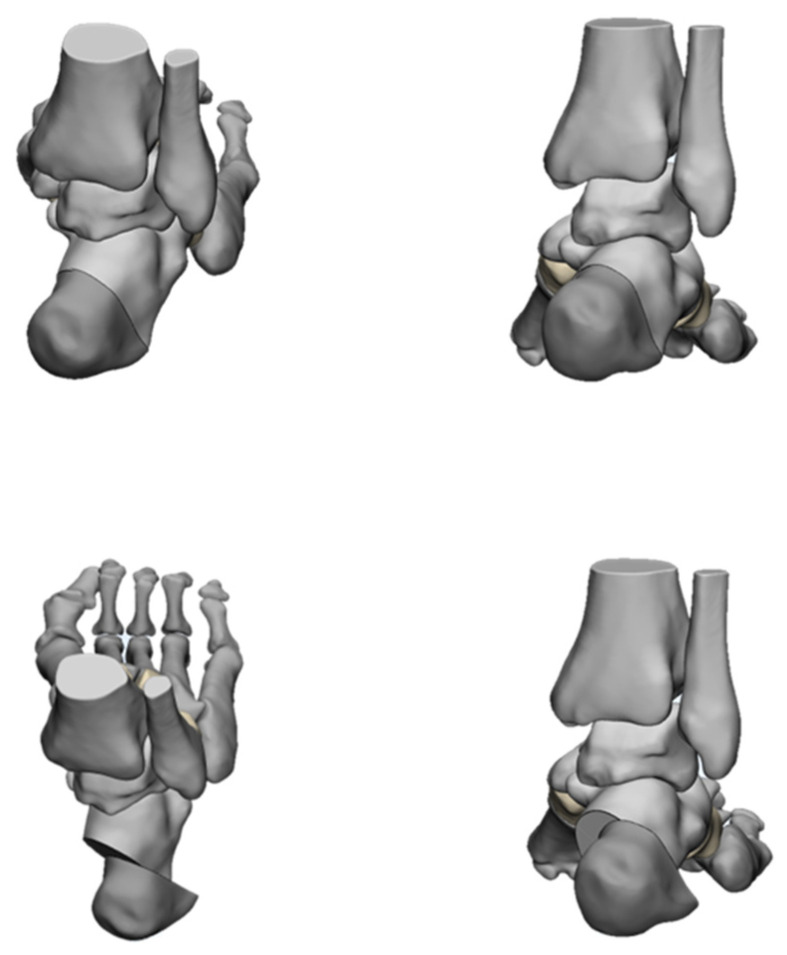
The oblique calcaneus Osteotomy.

**Figure 17 jcm-11-03679-f017:**
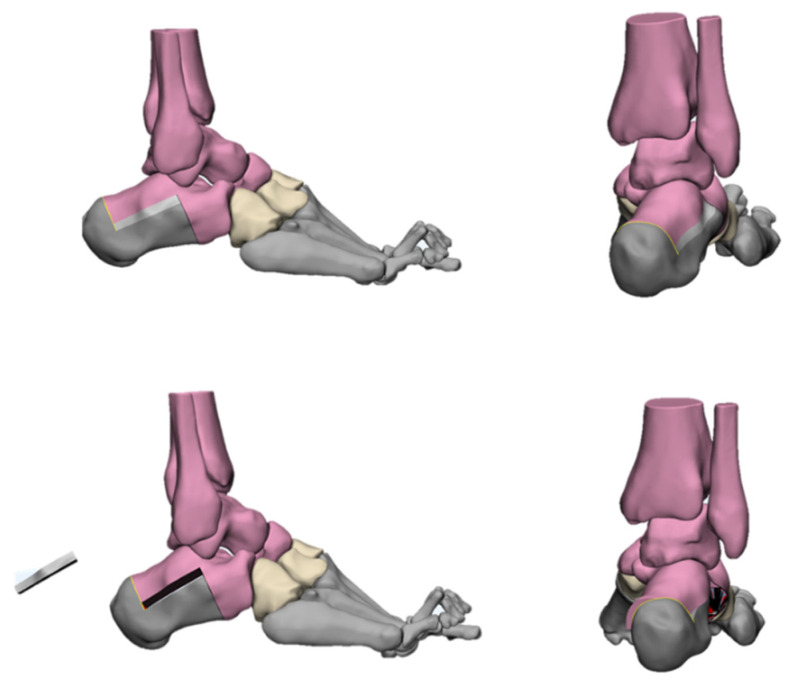
The Z-shaped calcaneus Osteotomy.

**Figure 18 jcm-11-03679-f018:**
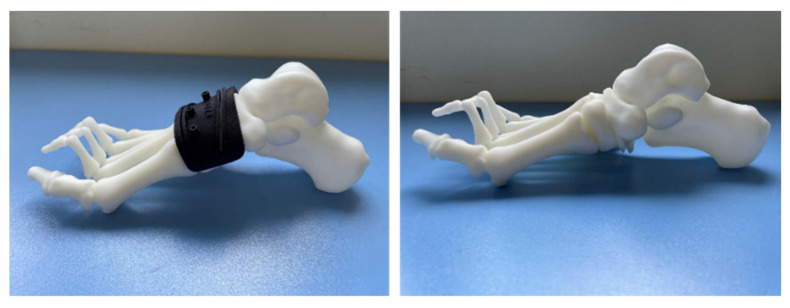
Three-dimensional printed midfoot osteotomy guide to assist cavus foot correction.

**Table 1 jcm-11-03679-t001:** Differential etiology of cavus foot deformity.

**Congenital**	Friedreich’s ataxia Traumatic brain injury Spina bifida Syringomyelia Filum terminale lipoma Tethered cord syndrome
Clubfoot Tarsal Coalition	
**Peripheral nervous system**	
Charcot-Marie-Tooth disease Guillain-Barre syndrome Peripheral neuropathy	
**Central nervous system**	**Traumatic**
Cerebral palsy Spinal cord tumor Stroke Amyotrophic lateral sclerosis Poliomyelitis Huntington’s chorea	Crush injury Postburn contracture Talar neck fracture malunion Peroneal nerve injury Crush injury

**Table 2 jcm-11-03679-t002:** Common associated manifestations of cavus foot.

**Forefoot**
Metatarsalgia
Stress fracture of the fifth metatarsal
Callus under first, fifth metatarsal heads
Claw-toes
Metatarsus adductus
**Midfoot**
Midfoot arthritis
Talar subluxation
**Hindfoot**
Plantar fasciitis
Achilles tendinitis
Subtalar unstable
Peroneal tendons subluxation
Peroneal tendon problems (tear or split, rupture, tendinopathy)
**Ankle**
Chronic lateral ankle instability
Varus ankle arthritis
**Gait**
Limp

**Table 3 jcm-11-03679-t003:** Chosen of surgical methods.

**Soft-Tissue Release**		
Contracture of the plantar fascia	→	Open or Percutaneous plantar fasciotomy
Overpull of the intrinsic muscle	→	Steindler stripping
Ankle varus deformity	→	Lateral ankle ligament reconstruction
		Deltoid ligament release
Ankle equinus deformity	→	Gastrocnemius recession
		Achilles tendon lengthening
Severe rigid deformity	→	Combined with other tendon release
**Bony Reconstruction**		
Forefoot deformity		
The first TMT equinus	→	First metatarsal dorsiflexion osteotomy
The multiple metatarsals equinus	→	Jahss osteotomy
Midfoot deformity		
The apex at the NC joint or cuneiforms	→	Cole/Japas/Akron/Myerson osteotomy *
		Ilizarov external fixation
Hindfoot deformity		
Nonreducible mild heel varus	→	Dwyer osteotomy
Nonreducible severe heel varus	→	Z-shaped osteotomy
Mixed deformity		
Rigid deformity with osteoarthritis	→	Double or Triple arthrodesis
		Naviculocuneiform arthrodesis
Soft-tissue Balancing		
Weakness of the peroneus brevis	→	Peroneus longus to brevis transfer
Overpower of the posterior tibial tendon	→	Posterior tibial tendon transfer
The claw-toes	→	Jones procedure
		Hibbs procedure

***** Indicates that the various midfoot osteotomy methods are not described in detail here. TMT: tarsometatarsal; NC: naviculocuneiform.

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
