# Peer review of "Evaluation and Management of Cavus Foot in Adults: A Narrative Review"

_jcm, 2022, doi:10.3390/jcm11133679_

Round 1

Reviewer 1 Report

Thank you for allowing me to review the article Evaluation and management of cavus foot in adults. The study discussed to provide a new insight about cavus foot, including anatomy, etiology, classification, pathophysiology, physical and radiology examination and managements in patients with cavus feet.

 The authors conducted an poor  and low careful review of the literature and not following standard bibliography evaluation guidelines.  The conclusion of this review is the evaluation and management of cavus foot is challenging for podiatrists all over the world.

It is an original work, very ambitious and of great importance for the  physician and to podiatrist.

The title should be adjusted to the content of the work and a type of systematic review.

I think that the abstract should be a bit more adjusted to the objective, methods, results and conclusion. This needs to be made clearer throughout the paper

The introduction section did not provide a clear rationale for carrying out the study (for example, why is your research question important? What gap in the literature is the study addressing?
I suggest in this section should be improved, with more information related with the quality of life and prevalence in this population.

  In the methodology, it is necessary to include the search strategy, including the terms used, datas. Also, this section have to describes in detail the inclusion and exclusion criteria.

Also, it is necessary to include the number of titles and abstracts reviewed, the number of full-text studies retrieved, and the number of studies excluded together with the reasons for exclusion. This information have to presented in a table. 

Furthermore, the discussion is the most interesting section of a systematic review. Please to include this section in this manuscrit 

In sum, although many issues would be resolved by careful proofreading, the manuscript needs a careful evaluation of data quality beyond applied methodology.  In other words, do we have enough data to arrive to strong conclusions?

Author Response

Dear Reviewer

Thank you for giving us a chance to revise and improve the quality of our article. Those comments are all valuable and very helpful.

We have read your comments carefully and have tried our best to revise our manuscript according to the comments. Attached please find the revised version and the supplement files, which we would like to submit for your kind consideration.

Reviewer 2 Report

Title and abstract

The aim of the review is unclear. Did the author search for current evidence? How?

Line 13: “management of patients”, not managements in patients

Introduction

- Lines 26-27: irrelevant and should be deleted

Anatomy, Etiology and Classification of cavus foot

- The anatomy section is better called “biomechanics”

- The solid foot model is not well described. 

Clinical evaluation

Silfverskiold test should be revised as the interpretation of the test is reversed between GG and Achilles.

What’s new?

This section should be written in detail.

Conclusions

Unclear. What are the new insights that the authors stated previously?

What does this review add to the current evidence?

Tables and Graphics

- Figure 2: It is better to state a cavovarus foot in the legend. CMT should be deleted, better not to use abbreviations inn the figure legend.

- Table 2: It is better named: “common associated manifestations of cavus foot”. Also what is the difference between unstable and limp?

The authors investigated an important topic. However, to make this manuscript publishable, they should address all my comments.

Author Response

Dear Reviewer

It’s our great honor to receive the comments on my paper from you. Those comments are all valuable and very helpful for revising and improving our paper.

     We have studied the comments carefully and have made corrections which we hope meet the approval. Revised portions are marked in red in the paper. You can find them in the attached files uploading. If there are any errors, please notify us. We will correct them accordingly.

     Once again, thank you very much for your comments and suggestions.

Reviewer 3 Report

Very nice paper. A lot of hard work. You should be congratulated. I think it will benefit CMT patients and their surgeons. Please see my comments below.

I’ve done more CMT surgeries than anyone in the United States, and we’ve published 7 scientific papers on CMT over the past 6 years. Please look in Foot and Ankle International, and look up articles by either myself, Glenn Pfeffer, or my co-author, Max Michalski, or Tonya An. I suggest including these papers in you work, as they are the most current research papers on the CMT topic.

One of the papers was authored with seven past presidents of the AOFAS, among others. It is a consensus statement on the treatment of CMT. The paper was sponsored by the Charcot Marie Tooth Association. Please modify your comment in the Abstract that “There are no standers or guidelines for the treatment of cavus .... until now.” That comment is not true.

In the introduction, you state that cavus is one of the most common problems .... Is that true in China? It certainly is not true here. Please comment on the prevalence of CMT in China. I know of no published data on that topic. Do you see CMT patients frequently?

Please look at sentences 28 - 33. Are your comments about gait correct? To me they are confusing.

Perhaps the most important, and only reliable radiographic measurement of the CMT foot, is overcoverage of the TN joint. You might discuss that. Perhaps the most important papers on CMT imaging were just published by us, one by Michalski and the other by An in FAI. Please include them in your paper.

I find it very rare to need a midfoot osteotomy. You seem to indicate it is frequently needed. If it is done, the Cole is by far the best and most widely accepted in the United States. Please refer to our consensus statement in FAI.

Please review our study by Michalski on the 3D imaging of calcaneal osteotomies. The Z osteotomy although written about in the literature is an awkward osteotomy that has little, if any indication. An in-depth discussion of heel osteotomies would be interesting for your paper. More interesting I think than the discussion of midfoot osteotomies.

Please consider this comment: Start surgery with an Achilles or gastroc to take the VARUS pull off of the calc. Then go to all soft tissue releases, including spring ligament and midfoot joints. ONLY THEN, does the surgeon know what osteotomies have to be done. Correct the heel first, if it is needed, and then move forward. Dividing the PF in the midfoot has the most power in freeing up a severely depressed first met. I’m not sure the Steindler, which I did in the past, does very much.

Do you transfer the PT tendon below the reticulum? Why go into the middle cuneiform if the lateral is central?

In your conclusion, why do you only refer to Podiatrists, on line 452? Orthopaedic surgeons don’t do CMT surgery in your country?

Once again, congratulations on a job well done. Update your bibliography and review our articles in Foot and Ankle International, and you will have an eminently publishable piece.

Author Response

Dear Dr. Glenn Pfeffer

It’s my great honor to receive the comments on my paper from you, the most famous foot and ankle surgeons all over the world. What a surprise!

 I am currently studying for a Ph.D. in Foot and Ankle surgery and am a beginner. Therefore, my understanding of cavus foot is not deep enough.

     I have read your bookOperative Techniques: Foot and Ankle Surgery” previously and benefited a lot. My supervisor, Dr. Zhang Hui, participated in the fellowship program of AOFAS in 2019.

     I have carefully read the scientific papers on CMT published by you and your team. You have done a lot of work on cavus foot. You set a good example for us.

     I have updated the bibliography and revised the paper according to your suggestions. Revised portions are marked in red in our paper. You can find them in the attached files uploading. If there are any errors, please notify me. I will correct them accordingly.

Reviewer 4 Report

In the manuscript, the author mainly summarized the current research involved in the Cavus foot. They provide sufficient information related with the anatomy, etiology, classification,  and pathophysiology for the Cavus foot. Moreover, they discussed how to diagnose and treat the patients with Cavus foot. These could help us better understand the underlying mechanisms and promising therapeutic strategy for the patients with Cavus foot. Overall, this review is well-organized, and the content is written in a logic pattern.  Following is some comments and suggestions.

(1) For the cavus foot, how about the racial differences for the disease?

(2) For the Fig.1, the author indicate that The TC (talo-calcaneal) angle become narrow. Could the author label the angle in the Fig.1? It could help the reader to easy understand and follow. 

(3) For the section 6 what is new, it is, the future direction of cavus foot research. Could the author provide the details for the most recent literatures? And how these 3D printed osteotomy guide plates could assist the correction of Cavus foot? It is interesting and a well-deserved topic to discuss. 

Author Response

Dear Reviewer

On behalf of my co-authors, we thank you for giving us a chance to revise and improve the quality of our article. Those comments are all valuable and very helpful.

We have read your comments carefully and have made revisions which were marked in red in the paper. We have tried our best to revise our manuscript according to the comments. Attached please find the revised version, which we would like to submit for your kind consideration.

Round 2

Reviewer 1 Report

  My opinion about the article remains the same of the first revision of manuscript. Most of the issues that I advanced to you cannot be repaired. A new study would be needed to make these things suitable. The clarifications provided do not solve the problem. Best regards.

Author Response

Dear Reviewer:

    Thank you for your valuable advice. In the future, I will conduct a systematic review or meta-analysis on a specific aspect of cavus foot. Best regards.

Reviewer 2 Report

Authors adequately addressed my suggestions for corrections.

Author Response

Dear Reviewer:

    Thank you for your valuable advice. Best regards.